# Magnetic Fe_2_O_3_–SiO_2_–MeO_2_–Pt (Me = Ti, Sn, Ce) as Catalysts for the Selective Hydrogenation of Cinnamaldehyde. Effect of the Nature of the Metal Oxide

**DOI:** 10.3390/ma12030413

**Published:** 2019-01-29

**Authors:** Robinson Dinamarca, Rodrigo Espinoza-González, Cristian H. Campos, Gina Pecchi

**Affiliations:** 1Depto. Físico-Química, Facultad de Ciencias Químicas, Universidad de Concepción, Edmundo Larenas 129, Concepción 4070371, Chile; robidinamarca@udec.cl (R.D.); ccampos@udec.cl (C.H.C.); 2Deparment of Chemical Engineering, Biotechnology and Materials, FCFM, Universidad de Chile, Beauchef 851, Santiago 8370456, Chile; roespino@ing.uchile.cl; 3Millenium Nuclei on Catalytic Processes towards Sustainable Chemistry (CSC), Santiago 8340518, Chile

**Keywords:** nanocatalyst, core shell, hydrogenation

## Abstract

The type of metal oxide affects the activity and selectivity of Fe_2_O_3_–SiO_2_–MeO_2_–Pt (Me = Ti, Sn, Ce) catalysts on the hydrogenation of cinnamaldehyde. The double shell structure design is thought to protect the magnetic Fe_2_O_3_ cores, and also act as a platform for depositing a second shell of TiO_2_, SnO_2_ or CeO_2_ metal oxide. To obtain a homogeneous metallic dispersion, the incorporation of 5 wt % of Pt was carried out over Fe_2_O_3_–SiO_2_–MeO_2_ (Me = Ti, Sn, Ce) structures modified with (3-aminopropyl)triethoxysilane by successive impregnation-reduction cycles. The full characterization by HR-TEM, STEM-EDX, XRD, N_2_ adsorption isotherm at −196 °C, TPR-H_2_ and VSM of the catalysts indicates that homogeneous *core-shell* structures with controlled nano-sized magnetic cores, multi-shells and metallic Pt were obtained. The nature of the metal oxide affects the Pt nanoparticle sizes where the mean Pt diameter is in the order: –TiO_2_–Pt > –SnO_2_–Pt > –CeO_2_–Pt. Among the catalysts studied, –CeO_2_–Pt had the best catalytic performance, reaching the maximum of conversion at 240 min. of reaction without producing hydrocinnamaldehyde (HCAL). It also showed a plot volcano type for the production of cinnamic alcohol (COL), with 3-phenyl-1-propanol (HCOL) as a main product. The –SnO_2_–Pt catalyst showed a poor catalytic performance attributable to the Pt clusters’ occlusion in the irregular surface of the –SnO_2_. Finally, the –TiO_2_–Pt catalyst showed a continuous production of COL with a 100% conversion and 65% selectivity at 600 min of reaction.

## 1. Introduction

Because of the versatility in composition and synthesis, materials with *core-shell* structure architecture have attracted particular attention in fields in which nanotechnology has a central role such as electronics [1], biomedicine [2], environmental remediation [3] and heterogeneous catalysis [4,5,6]. These materials in a nano- or micro-size structure produce a large surface-to-volume ratio, increasing the dominance of surface atoms and predominance of quantum effects, thereby improving their performance in various applications [7]. Efforts have been focused on the application of these materials in heterogeneous catalysts, specifically to increase their activity, operational stability and selectivity [8].

The selective catalytic hydrogenation of organic substrates containing a number of unsaturated functional groups is an important step in the industrial preparation of fine chemicals, increasing their relevancy for fundamental research in catalysis [9]. The heterogeneous selective hydrogenation, using H_2_ as reducing agent, of the carbonyl bonds in α,β-unsaturated aldehydes towards the formation of non-thermodynamically favored unsaturated alcohols, which are valuable intermediates in pharmaceutical, cosmetic, agrochemical and resin industries is still a challenge [10]. As a representative of α,β-unsaturated aldehydes, cinnamaldehyde (CAL) is principally used as a test molecule in the catalytic production of hydrocinnamaldehyde (HCAL), cinnamic alcohol (COL) and further hydrogenation to 3-phenyl-1-propanol (HCOL) [11], as shown in Scheme 1. In order to increase the selectivity of the COL production, different methods have been developed to improve the catalyst’s performance, such as screening the transition metals for active phases, supporters and promoters [12].

Among the transition metals, platinum (Pt) is the most-reported active phase in the hydrogenation of CAL. The selectivity of this active phase towards the formation of unsaturated alcohols has been attributed to the geometric and electronic defects of the metal [13] and can be improved by the incorporation of a second metal [14], a promoter [15], modification of the particle size [16] or by using a non-inert metal oxides in support [10,17] such as TiO_2_ [11], SnO_2_ [18], CeO_2_ [19], ZnO [20], among others. Moreover, the well-known strong metal-support interaction (SMSI) effect promoted by partially reducible metal oxide [21,22,23,24] is observed in metal-supported catalysts submitted to a reducing atmosphere at temperatures above 300 °C. At this condition, the metal oxide partially covers the metallic phase, promoting C=O catalytic hydrogenation performance.

In the field of heterogeneous catalysis, the use of *core-shell* catalysts as single unit with different components has been previously reported [25,26,27]. For CAL hydrogenation, Song et al. has reported the use of a Pt@CeO_2_ nanocatalyst achieving over 95% conversion with 87% selectivity to HCAL in 5 h under 1 atm H_2_ pressure [28]. In the same way, Liu et al. used MOF@Pt@MOF (MOF: Fe^III^-based MIL-100) as a catalyst to produce COL at 96% selectivity and 95% of conversion of CAL in 4 h of reaction. However, the preparation process of MOF catalysts is relatively cumbersome, uneconomical and inapplicable in mass production.

One way to produce improved Pt-based catalysts is to employ a *core-shell* configuration, including a magnetic core isolated by a layer of oxide shell, which could act as support for Pt-metal nanoparticles (Pt NPs). The magnetic *core-shell* structure catalysts could provide the additional property of easy removal of the nanostructured catalyst by an external magnetic field [28]. The magnetic particles can be synthesized by different chemical methods; the polyol process is an effective sol-gel route for nano- and micro-particle synthesis with a controlled shape and size [29]. Moreover, is possible to modify the core materials to produce single-shell, multi-shell, or porous-shell structures to improve catalytic activity, selectivity and structural stability [3]. An alternative approach for the protection of magnetic cores is the use of the Stöber method to produce silica-coated shells on magnetic NPs via the facile hydrolysis of tetraethyl orthosilicate (TEOS). It has been reported that uniform and robust silica shells have been formed by the complete hydrolysis of TEOS followed by the condensation of silicic acid, giving a network of tetrahedral SiO_4_ units with shared vertices [30].

The surface physiochemical properties of small crystallites change dramatically at sizes lower than 5.0 nm, where the metal or metal oxide particles have low-coordinated surfaces, improving the catalytic activity [31]. The development of core-multi-shell nano-supports allows the shell characteristics to be modified in a controlled way in terms of the particle size, shape or crystallinity to improve their catalytic effect in terms of activity, selectivity and operational stability.

The aim of this work is to provide new magnetic multifunctional Fe_2_O_3_–SiO_2_–MeO_2_–Pt (Me = Ti, Sn, Ce) nanomaterials as catalysts to be used in the selective hydrogenation of CAL. One of the most important steps to obtain the desired nanocatalyst structured microspheres is the coating process with TEOS for the formation of a silica shell as a protection layer for the magnetic core and a growth platform to provide for a homogeneous dispersal of a metal oxide shell on the Fe_2_O_3_–SiO_2_ surface. The synthesis of the catalysts, while employing stepwise impregnation-reduction deposition, provides highly dispersed Pt catalysts in a *core-shell* structure. The effect of the nature of the metal oxide MeO_2_ (Me = Ti, Sn, Ce) on well-stabilized magnetic core double shell structures in the selective hydrogenation of CAL was also described in this experiment.

## 2. Materials and Methods

### 2.1. Materials

Reagents were provided by Sigma^®^ (Darmstadt, Germany) and Merck^®^ (Darmstadt, Germany) Company and used without purification or treatment. These included iron chloride hydrate (FeCl_3_·6H_2_O), ethylene glycol, sodium acetate, poly(vinylpyrrolidinone) (PVP-K30), thetraethoxysilane (TEOS), ammonium hydroxide (NH_3_ 28 wt %), ethanol, tetrabutyl orthotitanate (TBOT), Cerium nitrate (III) hexahydrate (Ce(NO_3_)_3_·6H_2_O), tin(IV) tert-butoxide (TTB), urea, D-xylose, toluene, (3-aminopropyl)trimethoxysilane (APTMS), K_2_PtCl_6_ and NaBH_4_.

### 2.2. Fe_3_O_4_–SiO_2_ Structure

The Fe_3_O_4_–SiO_2_ structures were synthesized starting from Fe_3_O_4_–NPs coated with TEOS using the Stöber method in basic media [32]. The iron precursor (FeCl_3_·6H_2_O) was dissolved in an ethylene glycol and PVP-K30 mixture, and sodium acetate was added as a nucleating agent under vigorous magnetic stirring for 5 h. The solution was transferred to a Teflon-autoclave reactor (Hydrion Scientific Instruments Company Ltd., Baltimore, MD, USA) and heated at 200 °C for 8 h to produce a homogeneous and stabilized Fe_3_O_4_–NPs dispersion. Using magnetic separation, the dispersion was washed several times with an ethanol–water mixture before the coating procedure with Fe_3_O_4_–NPs and TEOS as silica precursor. Then, 2 mL of TEOS was added drop-wise to 300 mg of Fe_3_O_4_–NPs dispersed in a 180 mL ethanol, 60 mL water and 10 mL NH_3_ mixture, while stirring with mechanical agitation for 6 h at room temperature. The solid was separated by magnetization, washed three times with an ethanol–water mixture and dried at 50 °C for 12 h to produce Fe_3_O_4_–SiO_2_ nanostructures.

### 2.3. Fe_2_O_3_–SiO_2_ and Fe_2_O_3_–SiO_2_–MeO_2_ (Me = Ti, Sn, Ce) Structures

The coating process with titanium and tin was carried out with a modified Stöber method [33], with 300 mg of Fe_3_O_4_–SiO_2_ particles dispersed in 200 mL of ethanol and 1 mL ammonium solution, which was stirred for 1 h at 300 rpm under mechanical agitation. The condensation of the respective alkoxides was carried out by the addition drop-by-drop of 40 mL of ethanol with 5% by volume of the (Ti or Sn) alkoxide precursor to the dispersion and heating to 60 °C for 4 h. The coating process with Ce was carried out using a solvothermal methodology, as previously reported [34], with 300 mg of Fe_3_O_4_–SiO_2_ particles dispersed in 300 mL of absolute ethanol sonicated for 30 min at room temperature. The addition drop-by-drop of 60 mL of an aqueous dissolution of 1.2 g urea, 6.0 g D-xylose and 1.0 g cerium nitrate was carried out under mechanical agitation for 3 h. The mixture of the Ce precursor was transferred to a Teflon-autoclave reactor (Hydrion Scientific Instruments Company Ltd., Baltimore, MD, USA) and heated at 160 °C for 20 h. Finally, the Ti, Sn and Ce containing dispersions with Fe_3_O_4_–SiO_2_–shells were washed three times with ethanol, dried overnight at 50 °C and calcined at 500 °C for 6 h. For comparison purposes, Fe_3_O_4_ NPs and Fe_3_O_4_–SiO_2_ structures were also submitted to the calcination process at 500 °C for 6 h to produce Fe_2_O_3_ and Fe_2_O_3_–SiO_2_ structures.

### 2.4. Fe_2_O_3_–SiO_2_–MeO_2_–Pt (Me = Ti, Sn, Ce) Catalysts

To better-control metallic dispersion, the incorporation of Pt was carried out on functionalized APTMS structures. The functionalized procedure that was carried out contacted the Fe_2_O_3_–SiO_2_–MeO_2_ (Me = Ti, Sn and Ce) structures with APTMS under reflux in toluene for 24 h, was washed with a 3:1 mixture of acetone:toluene and dried at 50 °C for 12 h. Then, the deposition of Pt–NPs was carried out using an aqueous dissolution of K_2_PtCl_6_ as a precursor and NaBH_4_ as a reducing agent by successive impregnation-reduction steps until reaching a metallic loading of nominally ≈5 wt % Pt.

### 2.5. Characterization

The synthesis of the core-shell structures was studied by high-resolution transmission electron microscopy (HR-TEM) measurement. Morphology was examined using a FEI Tecnai G2 F20 S-Twin microscope (FEI, Hillsboro, OR, USA.), and the microanalysis was performed using a scanning transmission electron microscope (FEI, Hillsboro, OR, USA) and energy-dispersive X-ray spectroscopy (STEM-EDS) (FEI, Hillsboro, OR, USA). The total Pt and Fe content was determined by inductively coupled plasma-optical emission spectrometry (ICP-OES, Perkin Elmer optima 2100 DV ICP, (PerkinElmer, Waltham, MA, USA)) after dissolving the samples in 1:3 HNO_3_:HCl mixtures and diluting them with doubly distilled water. The surface and pore size distributions by nitrogen adsorption isotherms were obtained using a Micromeritics apparatus Model ASAP 2010 at −196 °C (Norcross, GA, U.S.A). X-ray powder diffraction (XRD) pattern measurements were carried out with nickel-filtered CuK_1_ radiation (λ = 1.5418 Å) collected on a Rigaku diffractometer (Rigaku, Tokyo, Japan) in the 2θ range of 10°–80°. Temperature programmed reduction (TPR-H2) profiles were obtained on Micromeritics TPR/TPD2900 equipment (Micromeritic, Norcross, GA, USA) provided with a thermal conductivity detector in 5% H_2_/Ar flow at a rate of 40 mL·min^−1^ from room temperature to 800 °C and a 5 °C·min^−1^ heating rate. The surface charges of materials were determined by zeta potential in a Zetasizer NanoZS Marvern instrument (Malvern, Worcestershire, UK) with a provided electrode tray. Magnetic measurements were carried out at room temperature in a vibrating sample magnetometer (VSM) Lakeshore 7400 series (Lakeshore, New Orleans, LA, USA) with a maximum applied field of 20 kOe. The VSM had been previously calibrated with a pure nickel sphere.

### 2.6. Catalytic Activity

The catalytic activity evaluation of cinnamaldehyde was performed in a batch compact Parr^®^ 5513 model reactor using 30 mg of catalyst weight and mechanical agitation up to 700 rpm. The reaction was performed at 100 °C, 2000 kPa of H_2_, with a substrate/catalyst mole ratio equal to 600 in a volume of 30 mL of cyclohexane. The reaction was monitored by taking non-invasive samples periodically from the reaction mixture at different reaction times until total cinnamaldehyde conversion. The product quantifications were carried out using a gas chromatograph (Hewlett Packard HP-4890, Palo Alto, CA, USA) equipped with a capillary column (HP-5) and a flame ionization detector (FID).

## 3. Results and Discussion

### 3.1. Material Synthesis

The different steps carried out to produce well-defined Fe_2_O_3_–SiO_2_–MeO_2_–Pt (Me = Sn, Ce, Ti) catalysts were carefully designed and the main stages are shown in Scheme 2. Fe_3_O_4_ nanoparticles were coated with TEOS to produce the starting core–SiO_2_ structure. The first layer of silica between the iron species in the core and the further deposition of the metal MeO_2_ (Me = Ti, Sn, Ce) oxide was designed to: (1) provide a regular, homogeneous and inert surface of SiO_2_ for metal oxide deposition, (2) inhibit the effect of the metal oxides changing the magnetic behavior of the iron species in the core and (3) evaluate the effect of the metal oxide in the catalytic performance of Pt without the effect of the iron species present in the core. Furthermore, the surface of the Fe_2_O_3_–SiO_2_–MeO_2_ (Me = Ti, Sn, Ce) structures were functionalized with APTMS before the impregnation of Pt, to enhance the metallic dispersion and to carefully control the Pt particle size with successive impregnation-reduction cycles.

### 3.2. HR-TEM

The morphology and composition of the core-shell structures during the synthesis was followed by HR-TEM and the corresponding EDX analysis. The Fe_3_O_4_–NPs (Appendix A) used as cores corresponded to micro-spheres with regular shapes formed by nanosize Fe_3_O_4_ particles joined together in intimate contact, forming aggregates of Fe_3_O_4_–NPs with a mean diameter of ~230 nm. Single Fe_3_O_4_–NPs have been reported to have regular square shapes of 20–50 nm [35] as well as 180-nm colloidal nanocrystal clusters of Fe_3_O_4_, each one composed of many single magnetite crystallites of 10 nm [36]. The partial reduction process of the Fe(OH)_x_ species to Fe_3_O_4_ is likely carried out by a dehydration process promoted by the presence of ethylene glycol (reducing agent) at 200 °C during the solvothermal synthesis [29]. These Fe_3_O_4_ nanoparticles are then coated by a first SiO_2_ shell by means of Stober’s method, generating a structure Fe_3_O_4_–SiO_2_
*core-shell* (Appendix A). This procedure allowed for the isolation of the magnetic core, thus providing a uniform surface to coat a second metal oxide shell. The mean diameter of ~320 nm for the Fe_3_O_4_–SiO_2_ structures allows for an average estimated thickness of ~50 nm silica.

The HR-TEM micrographs of Fe_2_O_3_–SiO_2_–MeO_2_ (Me = Ti, Sn and Ce) structures are shown in Figure 1a–c. As it was said before, during the calcination step to produce the –MeO_2_ oxides (Me = Ti, Sn and Ce), the Fe_3_O_4_ core was completely oxidized to γ-Fe_2_O_3_, in all of the samples, with almost no changes in the morphology. The Fe_2_O_3_–SiO_2_–TiO_2_ and Fe_2_O_3_–SiO_2_–CeO_2_ structures showed regular spherical shapes, which were indicative of a uniform metal oxide deposition on the surface of –SiO_2_. The Fe_2_O_3_–SiO_2_–SnO_2_ displayed an irregular surface attributable to the presence of larger SnO_2_ crystals. This trend is in agreement with the tin-based core-shell reported by Pang et al. [37], where the materials showed a non-defined coating after the deposition of SnO_2_ on the titanium dioxide core surface.

In Figure 1d–f the corresponding STEM-EDS is shown, where the analysis was carried out in the zone marked in Figure 1a by a red diametral line indicating the scan of the chemical analysis. The Fe_2_O_3_–SiO_2_–MeO_2_ (Me = Ti, Sn, Ce) structures appeared in the EDS signals. The presence of the respective metal oxides at the edges of the structures is indicative of a well-coated thin double shell. The thickness of the metal oxide shell is estimated by the differences in the average particle diameter of Fe_2_O_3_–SiO_2_ (Appendix A) in relation to Fe_2_O_3_–SiO_2_–MeO_2_ (Me = Ti, Sn, Ce) structures after counting more than 500 particles. The results obtained indicated a thickness of an average of ~20 nm for TiO_2_, ~30 nm for SnO_2_ and ~15 nm for CeO_2_. The similar thicknesses for TiO_2_ and SnO_2_ materials were a consequence of the reported modified-Stöber method [33,38] that favored the interaction between the precursor alkoxide with silica and the smallest thickness of the CeO_2_ coating materials added with the solvothermal method.

The HR-TEM micrographs of the Fe_2_O_3_–SiO_2_–MeO_2_–Pt (Me = Ti, Sn, Ce) catalysts are shown in Figure 2. All of the catalysts showed narrow particle size distributions, and their diameter increased in the order of Fe_2_O_3_–SiO_2_–CeO_2_–Pt < Fe_2_O_3_–SiO_2_–SnO_2_–Pt < Fe_2_O_3_–SiO_2_–TiO_2_–Pt. This effect is mainly attributable to the nature of the metal oxide during the Pt deposition. In the catalyst syntheses, the Fe_2_O_3_–SiO_2_–MeO_2_ (Me = Ti, Sn, Ce) were contacted with the metal precursor (K_2_PtCl_6_) dissolved in aqueous solution to deposit the active phase on the shells (SiO_2_, TiO_2_, SnO_2_ or CeO_2_ modified with APTMS). The core-shell materials can accumulate PtCl_6_^2−^ by a ligand-exchange process from the –NH_2_ surface group and/or adsorb the metal precursor on the surface of the metal oxide by electrostatic interaction. In Table 1, the zeta potential (ZP) measurements for the Fe_2_O_3_–SiO_2_–MeO_2_ (Me = Ti, Sn, Ce) aqueous dispersion are displayed. For all of the solids, ZP is a positive value, ascribable mainly to the presence of protonated amine groups from APTMS on the core-shell surface (–NH_3_^+^), because the *pK_a_* of APTMS is 10.6 [39].

Moreover, the metal oxides in the surface have different isoelectric point values (IEP): where IEP_TiO2_ = 5.9 [40], IEP_SnO2_ = 6.4–7.3 [41] and IEP_CeO2_ = 6.8–8.3 [40]. The IEP is a reference for predicting the charge-dependent behavior of oxide minerals and their suspensions, in which IEP is a zero-point charge arising from the interaction of H^+^ and OH^−^ with the solid and water [42]. In our case, the ZP at the pH of impregnation is positive, which enhances the metal precursors’ electrical attraction from the dissolution to the stationary layer of fluid attached to the dispersed particle. During the reduction process with NaBH_4_, the nucleation of PtCl_6_^2−^ to produce Pt NPs was enhanced in Fe_2_O_3_–SiO_2_–CeO_2_, mainly by the positive surface charge of –CeO_2_ at pH = 6.5. In the case of Fe_2_O_3_–SiO_2_–SnO_2_, the IEP of –SnO_2_ was close to the pH of the materials’ dispersion (see Table 1) which is in line with an increase in the Pt NPs after the reduction process in comparison with the –CeO_2_-based core-shell. For the Fe_2_O_3_–SiO_2_–TiO_2_, the IEP was lower than for the other solids, which provided a negative surface charge on –TiO_2_, which promoted an increase of Pt clusters during the reduction of Pt^4+^ to metallic Pt. Therefore, the large and homogeneous Pt metallic particle sizes can be attributed to the successive impregnation-reduction deposition. For comparison, we included HR-TEM micrograph for the Fe_2_O_3_–SiO_2_–TiO_2_–Pt, at 5 wt % Pt, prepared by the metal deposition in one step and a sequence of Pt depositions in multi-cycles for the Fe_2_O_3_–SiO_2_–TiO_2_–Pt catalyst (see Materials and methods section). The Fe_2_O_3_–SiO_2_–TiO_2_–Pt prepared in one step showed a mean Pt crystal size of 13.4 nm with a wide particle size distribution, which confirmed the irregular nucleation when this metal was loaded on the TiO_2_ shell surface during the Pt deposition. Moreover, the multi-step metal precursor addition provided for, in the first cycle, the nucleation of smaller Pt NPs, which grew uniformly in every cycle as shown in the Appendix A.

### 3.3. ICP

The elemental composition of the Fe_2_O_3_–SiO_2_–MeO_2_–Pt (Me = Ti, Sn, Ce) catalysts was determined by ICP spectroscopy and the results are in Table 1. A similar Fe content can be seen with almost no differences related to the metal oxide, which indicated a thin coverage of the double shell, as determined by EDX-Scan analysis. The Pt content approaches the nominal amount of 5 wt %, which was indicative of successful consecutive Pt impregnation-reduction cycles.

### 3.4. Specific Area

The adsorption–desorption nitrogen isotherms of the materials presented a type II IUPAC classification (Appendix A), which was indicative of non-porous structures. This result implied a non-textural modification during the sol-gel coating process not observed in the first shell of silica (see Table 1). It should be pointed out that, even though the materials had low surface areas, large platinum dispersions were observed in 5 wt % of Pt content catalysts, which supported the successful functionalization with APTMS and further Pt surface deposition impregnation-reduction cycles.

### 3.5. XRD

The XRD profiles of the Pt and non-Pt content structures are shown in Figure 3. Diffraction peaks can be seen, which indicated only the presence of maghemite γ-Fe_2_O_3_ (JCPDS 39-1356) as crystalline phases with an absence of Fe_3_O_4_ (JCPDS 65-3107) completely oxidized to Fe(III) (Appendix A). For the Fe_2_O_3_–SiO_2_–MeO_2_ (Me = Ti, Sn, Ce) structures, it can be seen that the coating process with the double shell of the metal oxide almost did not change the chemical composition of the crystalline Fe_2_O_3_ cores. There was only the apparition of segregated phases for the SnO_2_ structures, and the lack of detection of crystalline structures for TiO_2_ and CeO_2_ indicated that they were present with a high degree of dispersion (the crystal size of the oxides was lower than 5.0 nm). It has been reported for –TiO_2_ [4] that the nanocrystal size is controlled by the thickness of the TiO_2_ coating process. On the other hand, for –SnO_2_, the appearance of segregated phases of SnO_2_ rutile with a tetragonal crystal system [43] has also been reported. Therefore, the appearance of only SnO_2_ diffraction peaks is in agreement with the HR-TEM results, regarding a more heterogeneous second coating process attributable to the behavior of the tin precursor to form polycrystalline SnO_2_ nanoparticles [44].

With regard to the Pt-content catalysts, Fe_2_O_3_–SiO_2_–MeO_2_–Pt (Me = Ti, Sn, Ce) showed the same XRD profile as that of the pristine core-shell materials. This confirmed that the NaBH_4_ used to perform the reduction of the metal precursor had no effect on the structure of the Fe_2_O_3_–SiO_2_–MeO_2_ (Me = Ti, Sn, Ce). However, the diffraction line at 39.6°, which corresponded to the metallic Pt(111) with a face-centred cubic structure, was too weak to be used to determine particle size in the case of Fe_2_O_3_–SiO_2_–MeO_2_–Pt (Me = Ti, Sn, Ce). Indeed, it is well recognized that the XRD technique is limited by the size of the particles. In this case, the size of the particle is less than 5 nm, which was in agreement with the HR-TEM characterization (see Table 1).

### 3.6. TPR-H_2_

The TPR-H_2_ profiles shown in Figure 4, corresponded to values normalized to per gram Fe to compare the intensity of the TCD signals independent of the more easily reducible FeO_x_ species and large iron content. In order to better see the reduction profiles of the Fe_2_O_3_–SiO_2_–MeO_2_–Pt (Me = Ti, Sn, Ce) catalysts in Figure 4, the profiles of Fe_2_O_3_–NPs is also shown. The additional experiment to produce Fe_2_O_3_–NPs was carried out on non-coated Fe_3_O_4_–NPs submitted to an oxidation treatment. The Fe_2_O_3_–NPs profiles indicated a first reduction step at 370 °C from hematite Fe_2_O_3_ to Fe_3_O_4_, followed by a broad reduction process started above 400 °C that can be deconvoluted into two peaks attributable to the Fe_3_O_4_→FeO and FeO→Fe steps [45]. The similarity of the TPR profiles for the Fe_2_O_3_–SiO_2_–MeO_2_–Pt; (Me = Ti, Sn, Ce) catalysts to the Fe_2_O_3_–NPs profile was a consequence of the larger content of Fe_2_O_3_ compared to the TiO_2_, SnO_2_ and CeO_2_ oxides; the reduction process of TiO_2_, SnO_2_ and CeO_2_ was masked by the reduction process of Fe_2_O_3_. Unfortunately, the broadening of the peaks of Figure 4 (that could not be deconvoluted into different reduction steps) are likely unassignable to a particular reduction process. The absence of a reduction step at low temperature indicates that the deposition of the platinum nanoparticles reduced with NaBH_4_ at room temperature is the metallic state. To support this result, after the TPR of the –CeO_2_–Pt system, an oxidation process in air at 300 °C for 1 h was carried out before a second TPR profile (Appendix A). The reduction step at 100 °C in the second TPR confirms the presence of metallic Pt in the –CeO_2_–Pt catalyst. 

### 3.7. Magnetic Measurements

The magnetic hysteresis loops at room temperature of the materials scaled to g^−1^ of material are shown in Figure 5. The Fe_2_O_3_–SiO_2_–MeO_2_–Pt (Me = Ti, Sn, Ce) structures display a ferromagnetic behavior with a magnetic saturation of ~40 emu·µg^−1^. The remanence and coercivity properties of the hysteresis loop (inset of Figure 5) was attributable to the grain size of the core and limited for the silica coating process [2]. An increase in the coercivity of nanostructures has been reported due to an increase in both magnetocrystalline and shape anisotropy, which exert influence on the nanostructures’ magnetic properties [46]. Due to the metal oxide content, structures did not show a change in the shape of the hysteresis loop, which confirmed that the magnetic properties of the material were confined to the core structure. The observed remanence and coercivity values allowed for the easy separation of the prepared catalysts from the reaction medium using external magnetic fields [32]. This phenomenon is appreciated because it is possible to separate the solid from the middle of the reaction in 30 s after putting it in contact with a magnet (Figure 5).

### 3.8. Catalytic Performance

The selective catalytic hydrogenation of CAL to COL is an ambitious challenge and aspects such as the nanometric size (1–10 nm) of the active sites, surface structure, metal-support interactions and electronic effects become more important in achieving it [47], and these factors should be considered for describing a catalyst’s performance. To increase the selectivity towards the high-value-added COL, the nature of the shell as support for Pt NPs was studied as part of this work. The CAL conversion curves with reaction time for the Fe_2_O_3_–SiO_2_–MeO_2_–Pt (Me = Ti, Sn, Ce) catalysts are shown in Figure 6a, and to better illustrate the effect of the metal oxide type on the conversion levels, Figure 6b shows the successful fit of the experimental data with a pseudo 1st order reaction.

The calculated pseudo-1st-order reaction rate constants, the initial reaction rates, maximum conversions at 500 min. and selectivity are shown in Table 2. More than 99% conversion and 67% COL production selectivity were obtained at 300 min. when Fe_2_O_3_−SiO_2_−CeO_2_–Pt was used as catalyst, while the Fe_2_O_3_−SiO_2_−TiO_2_–Pt system only reached 78% conversion and 65% selectivity for COL during the same time of reaction. In contrast, the conversion decreased to 40% and the selectivity value lowered to 50% for the Fe_2_O_3_−SiO_2_−SnO_2_–Pt system.

For the selective hydrogenation of CAL, it is well-documented that the particle size of active metal species can exert vital impacts on their catalytic performance [47]. Generally, the decrease in the metal particle size leads to both the increased activity and the decreased COL selectivity. This is due to more active sites on small-sized metal particles that have less-coordinated surface sites and the higher adsorption strength of C=C bonds versus C=O bonds. In the other way, the use of –TiO_2_, –SnO_2_ and –CeO_2_ as shells was designed with the aim to increases the selectivity towards COL product, due to they have been reported as non-inert supports which could provide SMSI effect even at room temperature [23].

In the present work, although the sizes of Pt NPs were different (see Table 1), Fe_2_O_3_–SiO_2_–CeO_2_–Pt had the best catalytic performance. Due to the different sizes of Pt particles, there was a size dependence of hydrogenation activity among these catalysts. However, the pseudo-1st-order reaction rate constant follows the trend k_–CeO2_ > k_–TiO2_ >> k_–SnO2_, which is different from the trend for the Pt mean particle size, where –CeO_2_–Pt > –SnO_2_–Pt > –TiO_2_–Pt. This behavior can be explained by the nature of the –SnO_2_ shell in the Fe_2_O_3_–SiO_2_–SnO_2_–Pt. The –SnO_2_ shell possess an irregular surface with evident formations of SnO_2_ clusters with particle sizes larger than the Pt NPs, as was demonstrated by HR-TEM and XRD characterization. Despite the production of Pt NPs with a mean size of 3.5 nm (see Table 1), the crystals were almost randomly dispersed in the –SnO_2_ interstices as hetero-aggregates, which could block the active sites, thereby decreasing the conversion levels. In relation to the results for k_–CeO2_ > k_–TiO2_, the ratio of the pseudo-1^st^-order reaction rate constant was k_–CeO2_:k_–TiO2_ = 2.2:1, which is in agreement with the differences in the Pt particle sizes, where their size ratio in –TiO_2_–Pt:–CeO_2_–Pt was 1.9:1.

The selectivity was the most interesting result; Fe_2_O_3_–SiO_2_–TiO_2_–Pt catalysts showed a high selectivity to the desired product during the entire reaction course. For a better understanding, Figure 7 shows the product distribution as a function of time for all of the catalysts until 500 min. of reaction. The decrease of the catalyst selectivity in the Fe_2_O_3_–SiO_2_–CeO_2_–Pt system is in line with work reported by Wei et al in the selective hydrogenation of CAL over a Pt/CeO_2_ catalyst [19]. The Fe_2_O_3_–SiO_2_–CeO_2_–Pt catalyst displayed a sharp decrease in the CAL concentration in the initial reaction period. The CAL concentration reached near total consumption after 240 min. Meanwhile, the COL concentration increased gradually in the total reaction period of 120 min. However, the production of HCOL also increased rapidly after 120 min. of reaction, reflecting a thermodynamically favorable hydrogenation of the C=C bond. Despite of the presence of CeO_2_ which could promote the SMSI effect at the reaction temperature, as was observed at lower conversion level, the low mean Pt crystal size enhance the over-hydrogenation of both COL and HCAL producing mainly HCOL at high conversion level.

In the selective hydrogenation of CAL, it has been reported that the adsorption of the C=O bond towards the formation of the unsaturated alcohol is favored on larger metallic particle sizes [47,48,49,50]. However, for the Fe_2_O_3_–SiO_2_–TiO_2_–Pt catalyst, a continuous increase in the COL formation can be seen with no further second hydrogenation step. We propose that the chemical properties of the TiO_2_ loading shell could generate an active site favoring the adsorption through the C=O, which inhibited the coplanar adsorption of CAL by both Pt crystal size and SMSI effect. Moreover, in the Fe_2_O_3_–SiO_2_–TiO_2_–Pt catalyst, the total hydrogenation product (HCOL) results from the HCAL hydrogenation with non-consumption of COL at 180 min which are in agreement with the preferred C=O adsorption on the catalyst surface.

Finally, for the Fe_2_O_3_–SiO_2_–SnO_2_–Pt catalyst, the competitive adsorption of the substrate by the SnO_2_ crystallites and the homogeneous and narrow Pt particle sizes produced HCAL and COL in almost the same proportion. Moreover, although CAL hydrogenation does not require acid sites, in the smallest Pt nanoparticles, the Lewis acid properties of SnO_2_ influenced the reaction rate and the selectivity of the process by modifying the adsorptive properties of Pt because of surface electron density [50]. Therefore, the observed selectivity results can be attributed to surface modification and metal-support interactions as well as a decrease in the Pt particle size [47].

## 4. Conclusions

Fe_2_O_3_–SiO_2_–MeO_2_–Pt (Me = Ti, Sn, Ce) magnetic *core-shell* catalysts were successfully prepared. The use of –SiO_2_ shells provided a protected magnetic core, and its structure ensured the deposition of a second metal oxide shell, which preserved the shape of the Fe_2_O_3_–SiO_2_ structure. The Pt metal nanoparticles deposited in continuous cycles provided support to the catalyst with a homogeneous metal dispersion on the –MeO_2_ surface (Me = Ce, Sn, Ti). In addition, the nature of the metal oxide affected the Pt mean particle diameters in the order of –TiO_2_–Pt > –SnO_2_–Pt > –CeO_2_–Pt. This trend was attributed to the IEP of the solids, which affected the interaction of metal precursors with the surfaces of the metal oxide shells. Consequently, the catalytic activity showed the trend of –CeO_2_–Pt > –TiO_2_–Pt > –SnO_2_–Pt, which is line with the physicochemical properties of the core-shell catalysts. In the case of –CeO_2_–Pt, an excellent activity and a continuous decrease in the production of COL was attributed to the Pt mean particle size (2.6 nm), which enhanced the consumption of both CAL and COL to produce of HCOL at a higher CAL conversion. In the opposite direction, –SnO_2_–Pt showed a minor activity in comparison with the other catalysts, mainly attributed to the irregular SnO_2_ shell surface, which allowed for the formation of Pt nanoparticles inside the SnO_2_ crystals, thereby blocking the active phase, which decreased the catalytic performance. The large stability towards the production of the unsaturated cinnamic alcohol of the –TiO_2_–Pt catalyst is attributed to particle size (4.9 nm) and the chemical properties of the TiO_2_ loading shell favoring the adsorption of CAL by the C=O bond, inhibiting the coplanar adsorption of CAL to produce HCAL and/or HCOL. This study suggests both Pt nanoparticles size and SMSI effects have resulted beneficial in selective hydrogenation of CAL to produce COL as the main product.

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
