# Peer review of "Magnetic Fe2O3–SiO2–MeO2–Pt (Me = Ti, Sn, Ce) as Catalysts for the Selective Hydrogenation of Cinnamaldehyde. Effect of the Nature of the Metal Oxide"

_materials, 2019, doi:10.3390/ma12030413_

Round 1

Reviewer 1 Report

The authors synthesized magnetic Fe2O3@SiO2@MeO2-Pt (Me=Ti, Sn, Ce) catalysts and compared their catalytic performance of cinnamaldehyde hydrogenation in terms of selectivities and conversion rates. They observed the best selectivity toward cinnamic alcohol over TiO2 supported catalysts and attributed it to the "particle size and the chemical properties of the TiO2 loading shell favouring the adsorption of CAL by the C=O bond..." The authors emphasized the effect of Pt particle size while not considering other possible effects arise from the supports, e.g. metal-support interaction. Nonetheless, it is an interesting work and I would like to recommend it for publication after addressing the comments below. 

1) Strong metal-support interaction (SMSI) is a well-known interaction in reducible metal-oxide supported group VIII metal catalysts (e.g. see Tauster's seminal works, JACS, 1978, 100, 170 or more recent works related to this effect ACS Catal. 2016, 6, 2, 974). The authors should discuss the role of this interaction in the selectivities and activities of their catalysts. Even though the reaction temperature used in this study is lower than the typical temperatures to induce SMSI, still, under reducing environment for a long time, especially on CeO2 and probably on TiO2 as well, SMSI could be induced. The decreased selectivities over CeO2 could be well due to the SMSI effect induced during the reaction.

2) For the support of SnO2, how could the authors exclude the possibility of intermetallic PtSn nanoparticle formation, which would have drastically different catalytic properties compared to pure-phase Pt. XRD could differentiate Pt and PtxSny (see Angew. Chem., 2017, 56, 3925), however, the XRD data presented in the current paper could not detect Pt due to the low loading. High-resolution HAADF-STEM can be informative in this regard as Sn and Pt yield stark contrast (see Chem, 2018, 4, 1387) or perhaps EDS mapping would give some hint. 

Author Response

Please find attached the answers to the reviewers

Reviewer 2 Report

The paper presents an experimental study of preparing of Fe2O3-SiO2-MeO2-Pt core-shell structures for catalytic applications. The experimental investigation and analysis are well done and scientifically sound but some parts of paper need correction. Thus the paper is suitable for publication in Materials after the following points have been carefully addressed (minor revisions):

In a title, abstrac and in text  is Fe2O3@SiO2... It should be Fe2O3-SiO2. With "-" instead of "@".

The same in core@shel (e.g. line 21, 34, 83. ..).

Line 139: "was followed by hihg..." better will be "was investigated/studied".

Line 142 and 204: you wrote STEM-XEDS, but in line 204 is EDX signal. It should be EDX signal or STEM-EDS system.

Fig. 1 and Fig 2. Add a larger scale in TEM images.

Line 242. How you know, that is reduction of Pt ions? Did you perform XPS measurements?

Author Response

(The authors gave the same response as above.)

Round 2

Reviewer 1 Report

I think the authors should include the discussions of the possible effect of SMSI into the manuscript. Other than this, the manuscript is publishable.

Author Response

Based on the comments made by the referees regarding the SMSI effect, the authors have incorporated the following references and discussions of the results into the manuscript.

Line 58: “Moreover, the well-known strong metal-support interaction (SMSI) effect promoted by partially reducible metal oxide is observed in metal--supported catalysts  submitted to  a reducing atmosphere at temperatures above 300 °C. At this conditions, the metal oxide partially cover the metallic phase, promoting C=O catalytic hydrogenation performance.”

Line 242: “In the other way, the use of -TiO2, -SnO2 and -CeO2 as shells was designed with the aim to increases the selectivity towards COL product, due to they have been reported as non-inert supports which could provide SMSI effect even at room temperatura”

Line 367: “Despite of the presence of CeO2 which could promote the SMSI effect at the reaction temperature, as was observed at lower conversion level, the low mean Pt crystal size enhance the over-hydrogenation of both COL and HCAL producing mainly HCOL at high conversion level”

Line 375: “We propose that the chemical properties of the TiO2 loading shell could generate an active site favoring the adsorption through the C=O, which inhibited the coplanar adsorption of CAL by both Pt crystal size and SMSI effect”

Line 379: “….which are in agreement with the preferred C=O adsorption on the catalyst surface.”

Line 407: “This study suggests both Pt nanoparticles size and SMSI effects have resulted beneficial in selective hydrogenation of CAL to produce COL as the main product.”